# Microbiome Shifts and Their Impact on Gut Physiology in Irritable Bowel Syndrome

**DOI:** 10.3390/ijms252212395

**Published:** 2024-11-19

**Authors:** Ioanna Aggeletopoulou, Christos Triantos

**Affiliations:** Division of Gastroenterology, Department of Internal Medicine, University Hospital of Patras, 26504 Patras, Greece; iaggel@upatras.gr

**Keywords:** gut microbiota, microbiome, brain–gut axis, irritable bowel syndrome, IBS, pathophysiology, visceral hypersensitivity

## Abstract

Irritable bowel syndrome (IBS) is one of the most prevalent functional gastrointestinal disorders characterized by recurrent abdominal pain and altered bowel habits. The exact pathophysiological mechanisms for IBS development are not completely understood. Several factors, including genetic predisposition, environmental and psychological influences, low-grade inflammation, alterations in gastrointestinal motility, and dietary habits, have been implicated in the pathophysiology of the disorder. Additionally, emerging evidence highlights the role of gut microbiota in the pathophysiology of IBS. This review aims to thoroughly investigate how alterations in the gut microbiota impact physiological functions such as the brain–gut axis, immune system activation, mucosal inflammation, gut permeability, and intestinal motility. Our research focuses on the dynamic “microbiome shifts”, emphasizing the enrichment or depletion of specific bacterial taxa in IBS and their profound impact on disease progression and pathology. The data indicated that specific bacterial populations are implicated in IBS, including reductions in beneficial species such as *Lactobacillus* and *Bifidobacterium*, along with increases in potentially harmful bacteria like Firmicutes and Proteobacteria. Emphasis is placed on the imperative need for further research to delineate the role of specific microbiome alterations and their potential as therapeutic targets, providing new insights into personalized treatments for IBS.

## 1. Introduction

Irritable Bowel Syndrome (IBS) is a chronic functional gastrointestinal disorder characterized by abdominal pain or discomfort, bloating, and altered bowel habits [1,2]. IBS is classified into four distinct subtypes: IBS with predominant diarrhea (IBS-D), IBS with predominant constipation (IBS-C), mixed IBS (IBS-M), and unclassified IBS (IBS-U) [3,4]. IBS-U is used when a patient’s stool patterns do not consistently align with the criteria for the other subtypes, making classification more difficult [3,4]. The diagnosis of IBS can be challenging, as symptoms may not only fluctuate over time [5], but can also resemble those of other gastrointestinal disorders [6,7,8]. It is estimated that around 10-15% of the global population suffers from IBS, with prevalence rates ranging from 5% to 20% [9,10]. These rates vary based on geographical location, diagnostic criteria, and demographic factors such as age, gender, and lifestyle. Higher prevalence rates are typically observed in Western countries, while lower rates are reported in Asian countries [9,10]. IBS disproportionately affects females, with a prevalence 1.5 to 3 times higher than in males [11,12]. It can occur in individuals of all age groups, with 50% of cases presenting by the age of 35, and its prevalence decreases in those over 50 years old [11,12].

The absence of a definitive diagnostic test or biomarker for IBS resulted in the development of symptom-based diagnostic criteria by expert consensus, such as the Rome criteria, to standardize diagnosis and reduce unnecessary testing [13]. The criteria have been revised over time, with the Rome III criteria established in 2006 [14] and the Rome IV criteria implemented in 2016 [1]. Notably, the Rome IV criteria made key changes, including the exclusion of abdominal discomfort from the definition and increasing the threshold for abdominal pain from at least 3 days per month to at least 1 day per week [1]. Consequently, the Rome IV criteria are more restrictive than the previous version, resulting in fewer individuals who self-diagnose with IBS meeting the formal diagnostic criteria. However, those who do meet the criteria exhibit more severe symptoms and higher levels of psychological comorbidity [15,16].

IBS is closely related to a significant decline in health-related quality of life [17], as well as increased somatization rates [18], higher psychological comorbidities such as depression and suicidal thoughts [19], and work productivity impairment [20]. Additionally, individuals with IBS incur higher annual medical and prescription costs, and the condition contributes substantially to both primary and secondary healthcare expenditures [21,22]. Despite being diagnosed in primary care and following current guidelines, many IBS patients are still referred for colonoscopy, even though the diagnostic yield is low for those with typical symptoms [23]. This may be driven by diagnostic uncertainty, which is often exacerbated by the chronic nature of IBS and the lack of a definitive cure.

The pathophysiology of IBS, although not yet fully understood, is generally considered to be multifactorial [24]. Genetic and epigenetic factors, along with stress-related nervous and endocrine systems, immune dysregulation, and the brain–gut axis (BGA), are thought to play significant roles in predisposing individuals to IBS [25,26]. In addition to these predisposing factors, environmental triggers such as food, toxins, adverse life events, chronic infections, and alterations in the gut microbiota are thought to play a critical role in initiating IBS symptoms [25,26]. Growing evidence has focused on the complex interactions between alterations in the gut microbiota (dysbiosis), changes in gut motility, mucosal inflammation, and the central nervous system (CNS), particularly through mechanisms such as visceral hypersensitivity and the BGA [27,28,29,30]. This evidence suggests that the gut microbiome may influence the CNS and vice versa, contributing to the complex pathophysiology of IBS.

The aim of the current review is to critically examine how shifts in gut microbiota composition affect key physiological processes related to IBS, including gut motility, mucosal inflammation, gut permeability, immune activation, and the BGA. Our focus emphasizes the concept of “microbiome shifts”, highlighting the dynamic alterations in specific bacterial taxa that become enriched or depleted in IBS and their consequential impacts on disease pathology. By synthesizing recent advancements in microbiome research, we investigate the mechanisms through which microbiota imbalances may drive IBS symptoms, with particular emphasis on the role of these microbial shifts in promoting visceral hypersensitivity and other critical symptoms of IBS.

## 2. Gut Microbiome

The gut microbiome consists of a diverse array of microorganisms, including bacteria, viruses, fungi, and protozoa, which play crucial roles in various physiological processes [31]. These microbes contribute to the metabolism of dietary nutrients and medications, support the structural integrity of the gut mucosal barrier, modulate immune responses, and provide protection against harmful pathogens [32]. The proximal colon and the cecum harbor the largest amount of microbiota, while both the large and small intestines contain a similar microbial biomass. Although the composition of the microbiota experiences significant changes with aging [33], it tends to remain stable over time in a healthy colon [34]. Currently, only about one-third of bacterial species have been identified and characterized, with the predominant phyla in the gastrointestinal tract being Firmicutes, Bacteroidetes, Proteobacteria, Actinobacteria, and Verrucomicrobia [35]. However, recent advancements in technologies, including metagenomics and metatranscriptomics, have significantly enhanced our understanding of gut microbiota [36].

The commensal organisms that typically inhabit the gut play a critical role in regulating signaling molecules and metabolites essential for maintaining gut homeostasis and supporting the development of the mucosal immune system [37]. The close interaction between the gut microbiome and the immune system plays a critical role in maintaining immune balance. Even minor disruptions in the gut microbiome can lead to immune dysfunction, triggering inflammation and abnormal immune responses. Such disturbances may weaken the integrity of the intestinal barrier, resulting in increased intestinal permeability [38]. This condition permits harmful substances, such as toxins, bacteria, and other molecules, to pass into the bloodstream, which in turn triggers an inflammatory response and worsens symptoms associated with inflammatory and autoimmune disorders. This condition is known as gut dysbiosis. Factors contributing to dysbiosis include antibiotic use, dietary changes, lifestyle shifts, exposure to severe enteritis diet, stress, and genetic predisposition [38]. Dysbiosis can disrupt mucosal homeostasis and has been implicated in a range of gastrointestinal disorders, including IBS [39,40].

Small intestinal bacterial overgrowth (SIBO), a condition characterized by the excessive proliferation of colonic-type bacteria including *Klebsiella*, *Enterococcus*, and *Escherichia* species, is common in IBS patients [41]. SIBO may lead to augmented gut permeability, low-grade inflammation, impaired motility, altered bile salt absorption, and disrupted enteric nervous system (ENS) function [41]. Additionally, the overgrowth of enteric archaea, such as *Methanobrevibacter smithii*, has been linked to IBS symptoms [42]. This archaeon produces significant amounts of methane in the small intestine, which influences gut motility, slows bowel transit, and is potentially associated with IBS-C [42].

## 3. Gut Microbiome and IBS

Recent developments in microbiome research utilizing cutting-edge microbiological technologies, such as whole genome sequencing and multi-omics analysis, have revealed the role of dysbiosis in the pathophysiology of IBS [43,44]. Approximately 10% of IBS cases have been reported following episodes of gastroenteritis, leading to PI-IBS [45]. Notably, the severity of IBS symptoms has been inversely correlated with microbiome density [46]. While findings across various studies have been inconsistent, the majority of research indicates that IBS patients exhibit reduced bacterial diversity and increased temporal instability in their microbiota [47]. Recent research has identified a potential gut microbiome signature linked to severe IBS [48]; however, the characterization of the intestinal microbiome in IBS remains inconsistent, with no universally accepted distinct signature [49].

### Gut Microbiome Alterations in IBS: Results from Meta-Analyses

Numerous meta-analyses have investigated the IBS-related alterations in gut microbiome composition, revealing distinct patterns that may contribute to the condition’s pathophysiology (Figure 1).

A meta-analysis that included ΙΒS patients demonstrated an elevation in Firmicutes and a reduction in Bacteroidetes, resulting in a higher Firmicutes-to-Bacteroidetes ratio at the phylum level [50]. Additionally, changes were noted at more specific taxonomic levels, including increased concentrations of Clostridia and Clostridiales, while Bacteroidia and Bacteroidales were found to be decreased [50]. Another study demonstrated an increase in bacterial populations from the family Lactobacillaceae, the genus *Bacteroides*, and the family Enterobacteriaceae (phylum Proteobacteria) compared to healthy controls [51]. In contrast, levels of uncultured Clostridiales, the genus *Faecalibacterium*, and the genus *Bifidobacterium* were found to be reduced [51]. Additionally, microbial diversity in IBS patients was either diminished or showed no significant difference compared to controls [51]. Liu et al., in their meta-analysis, indicated a decline in *Bifidobacterium*, *Lactobacillus*, and *Faecalibacterium prausnitzii* among IBS patients [52]. Likewise, another meta-analysis found diminished levels of *Lactobacillus* and *Bifidobacterium* but elevated levels of *Escherichia coli* and *Enterobacter* in the gut microbiome of IBS patients compared to healthy controls [53]. Interestingly, no significant differences were observed in the levels of fecal *Bacteroides* or *Enterococcus* [53]. In a recent meta-analysis, the fecal bacterial profiles and dysbiosis index of patients with IBS and its subgroups were compared to those of healthy controls [54]. The findings regarding fecal bacterial markers were inconsistent; however, two studies consistently reported elevated levels of *Ruminococcus gnavus* in IBS-D patients compared to healthy controls [54]. Additionally, the dysbiosis index indicated that patients with IBS and its subgroups, particularly those with IBS-D, exhibited higher dysbiosis index values than controls [54].

These meta-analyses reveal a consistent pattern of dysbiosis in IBS patients, characterized by an altered Firmicutes-to-Bacteroidetes ratio, reduced beneficial bacteria (e.g., *Bifidobacterium*, *Lactobacillus*, *Faecalibacterium*), and increased pro-inflammatory or potentially pathogenic bacteria (e.g., Enterobacteriaceae, *Escherichia coli*). The changes in microbial composition, particularly in IBS-D, suggest that these alterations may contribute to the condition’s pathophysiology, potentially exacerbating symptoms through immune activation, inflammation, or altered gut permeability. However, there is variability in specific bacterial markers across studies, and more research is needed to fully elucidate the role of these microbial shifts in IBS.

Lastly, scientific interest has expanded to include other components of the gut microbiota. Fungal populations, particularly *Saccharomycetes* and *Candida* species, have been found to differ in individuals with IBS compared to healthy controls, showing distinct genotypic and phenotypic variations [55]. However, the full implications of these alterations in the gut mycobiota are not yet fully understood. Additionally, advancing knowledge about the roles of archaea, viruses, phages, and protozoa could significantly shift our understanding of the microbiota in the future [56].

## 4. Brain–Gut Axis and Gut Microbiome

A central aspect of gut health lies in the intricate bidirectional communication between the gut and brain. This interaction not only regulates gut motility and sensitivity but also modulates emotional and pain responses, making it crucial to understanding gut disorders, like IBS. The BGA connects the CNS and ENS through endocrine, neural, and immune pathways, with the gut microbiota playing a crucial role in regulating this interaction [57]. Research has shown that bacterial colonization is essential for the normal development of both the enteric and central nervous systems [58]. The intestinal microbiome is in close contact with the CNS to maintain gut homeostasis and contributes to the production and regulation of neurotransmitters, immunomodulation, and the integrity of the intestinal barrier [58]. It also influences nociceptive sensory pathways involved in visceral pain, as well as gut permeability and motility [58]. Additionally, the secretion of neurotransmitters in response to stress has been associated with the growth of pathogenic bacteria like *Pseudomonas aeruginosa* and *Campylobacter jejuni* [58].

In parallel, evidence in germ-free mice has demonstrated that commensal bacteria are crucial for the development and maturation of CNS and ENS, as well as for regulating stress responses and anxiety-like behaviors [58,59]. Gut bacteria also regulate the serotoninergic system, with increased (5-HT) serotonin turnover observed in germ-free models [60,61]. For instance, *Bifidobacterium dentium* and its metabolite were found to enhance intestinal serotonin levels and serotonin receptor expression in animal models while also reducing anxiety-like behaviors [62]. Luck et al. demonstrated that *Bifidobacterium dentium* is capable of producing gamma-aminobutyric acid (GABA) from glutamate, glutamine, and succinate [63]. This was confirmed through genome analysis, which revealed the necessary genes for GABA synthesis, and by the observation of increased fecal GABA levels in *Bifidobacterium dentium*-mono-associated mice compared to germ-free control mice [63]. This finding highlights the role of gut bacteria in modulating neurotransmitter levels in the body [63]. Additionally, *Bifidobacterium dentium* was able to synthesize tyrosine but did not produce L-dopa, dopamine, norepinephrine, or epinephrine [63]. In vivo, *Bifidobacterium dentium* mono-associated mice showed elevated tyrosine levels in both feces and brain tissue, suggesting that this microbe highly contributes to neurotransmitter regulation [63]. Additionally, *Bifidobacterium adolescentis* produce GABA, further highlighting the role of the gut microbiota in modulating the BGA and related behavioral responses [64]. In a functional study, mice receiving IBS-D fecal microbiota had a similar microbial composition to controls but exhibited distinct metabolomic profiles, faster gastrointestinal transit, gut barrier dysfunction, immune activation, and anxiety-like behavior [65]. In a placebo-controlled trial, *Bifidobacterium longum* improved depression and quality of life in IBS patients, but not anxiety scores, using brain magnetic resonance imaging showing reduced responses to negative stimuli [66]. Table 1 illustrates the diverse roles of microorganisms, both pathogenic and beneficial, in modulating the BGA, influencing neurotransmitter production, immune response, and behavioral outcomes relevant to IBS pathology.

## 5. Post-Infection IBS

Disruptions in the BGA can become more pronounced following an acute infection. Chronic gastrointestinal symptoms following episodes of gastroenteritis have been documented for many years [67,68]. Since then, numerous studies have consistently described post-infectious IBS (PI-IBS), with incidence rates varying from 3% to 31% [69]. A comprehensive meta-analysis tracking individuals after acute gastroenteritis revealed that around 14.5% developed IBS more than a year later [70]. Factors that increase the risk of PI-IBS include female gender and pre-existing psychosomatic conditions [70]. The severity of the initial gastroenteritis, indicated by symptoms like prolonged duration (greater than seven days) or the presence of blood in the stool, also correlates with a higher likelihood of developing PI-IBS [70].

Various pathogens and disruptions in the microbial community of the gut contribute to the onset of IBS and worsening of symptoms. Various pathogens, including *Escherichia coli*, *Clostridium difficile*, *Campylobacter concisus*, *Helicobacter pylori*, *Giardia lamblia*, *Chlamydia trachomatis*, *Mycobacterium* avium subspecies *Paratuberculosis*, *Salmonella* spp., *Campylobacter jejuni*, *Pseudomonas aeruginosa*, and *Shigella* spp., have been implicated in exacerbating IBS [71]. *Campylobacter* infections, in particular, have been shown to impair the intestinal epithelial barrier, leading to increased gut permeability and cellular damage [72]. These infections trigger an inflammatory response characterized by the activation of various immune cells, including T-lymphocytes, macrophages, and mast cells, and the release of pro-inflammatory cytokines [73]. This immune response significantly disrupts processes such as vascular permeability, pain signaling pathways, gastrointestinal motility, and secretion, ultimately contributing to the clinical presentation of IBS.

The complex interplay between infection and stress and the development of PI-IBS has been explored. Psychological stress has been shown to exacerbate the effects of infection, increasing the risk of developing colonic inflammatory responses and worsening gastrointestinal disease [74], which in turn raises the likelihood of IBS development. When individuals experience stress during or shortly before an infection, the risk of developing PI-IBS is heightened [75]. This suggests that stress may prime the gut and CNS, making them more susceptible to the impacts of the infection, such as changes in gut motility and increased gut permeability [76]. Additionally, stress can influence the gut’s barrier function, disrupting the intestinal epithelial lining; this disruption may lead to intestinal permeability, allowing toxins and pathogens to cross into the bloodstream, triggering inflammation and immune responses that are associated with IBS symptoms [77]. Furthermore, a compromised gut barrier might lead to an increased intestinal microbial dysbiosis, which has been implicated in the pathophysiology of IBS [78]. Lastly, the CNS and the ENS, which control the functions of the gut, are deeply interconnected. Stress activates the hypothalamic–pituitary–adrenal (HPA) axis, increasing the release of stress hormones like cortisol, which can affect both the ENS and gut barrier function [79,80]. Moreover, the interaction between stress and gut neurons can lead to visceral hypersensitivity, a hallmark of IBS, by enhancing the responsiveness of the gut’s sensory neurons to stimuli.

Table 2 summarizes how pathogenic microorganisms contribute to PI-IBS by disrupting gut permeability, triggering inflammation, and altering gut motility.

## 6. Mucosal Immune Activation and Gut Microbiome

Mucosal immune activation in IBS is characterized by low-grade inflammation and infiltration of immune cells, particularly mast cells, into the intestinal mucosa. Mast cells, which express toll-like receptors 2 (TLR2) and TLR4, are activated by increased levels of fecal and serum lipopolysaccharides (LPS) in IBS, triggering the release of inflammatory mediators like histamine, tryptase, and prostaglandin E2 [81]. These mediators contribute to mucosal barrier dysfunction and visceral hypersensitivity, potentially leading to IBS symptoms. Alterations in the gut microbiome, including an increase in Bacteroidetes and a reduction in uncultured Clostridiales [82], have been observed in PI-IBS, correlating with increased mucosal expression of inflammatory cytokines, such as IL-1β and IL-6 [82]. In comparison to healthy controls, individuals with PI-IBS exhibit an augmented number of T lymphocytes, intraepithelial lymphocytes, mucosal enteroendocrine cells, enterochromaffin cells, and mast cells within the lamina propria [83,84,85]. These observations indicate that mucosal immune activation, triggered by disturbances in the gut microbiome, plays a key role in the pathophysiology of PI-IBS.

Cytokines play a crucial role as modulators in intestinal inflammation. Some cytokines, such as the anti-inflammatory IL-10, reduce the risk of developing IBS, while others are linked to its onset [86]. The key pro-inflammatory cytokines most commonly associated with IBS are IL-6, IL-8, and TNF-α [87]. Studies have reported elevated levels of IL-6, IL-8, IL-12, and TNF-α, along with reduced levels of the anti-inflammatory cytokine IL-10, in IBS patients compared to healthy individuals [88]. Probiotics and bacterial metabolites, such as butyrate from dietary fiber fermentation, have been shown to help restore cytokine balance and exert anti-inflammatory effects, potentially offering therapeutic benefits for IBS patients [89].

## 7. Intestinal Permeability and Gut Microbiome

One consequence of mucosal immune activation in a variety of gastrointestinal and non-gastrointestinal conditions, including IBS, is the disruption of the intestinal barrier, leading to increased intestinal permeability, or leaky gut. This breakdown allows harmful antigens to pass through, which can further escalate immune responses and contribute to gut dysfunction. Studies on the pathophysiology of IBS have highlighted the role of bacteria, particularly through fecal transplantation experiments. Transferring fecal matter from patients with IBS-C and IBS-D to germ-free animals has been shown to induce intestinal permeability, visceral hypersensitivity, and transit changes within the recipient animals [65,90,91]. Impairment of the gut barrier has been identified in patients with IBS-D, as well as those with PI-IBS, with prevalence rates ranging from 37–62% and 17–50%, respectively [92,93]. In contrast, IBS-C showed lower prevalence (4–25%), and no increased permeability was observed in IBS-M patients [92]. The loss of barrier function was linked to symptom severity, including abdominal pain, altered bowel habits, and psychological comorbidities [92].

The gut microbiome highly contributes to the development of IBS symptoms triggered by dietary factors. A high-FODMAP diet caused gut dysbiosis, increased fecal Gram-negative bacteria and LPS levels, and resulted in intestinal inflammation, barrier dysfunction, and visceral hypersensitivity in a rat model [94]. These effects were primarily mediated through an LPS-TLR4 signaling pathway, which reduced the expression of tight junction proteins in the colonic epithelium [94]. These effects were reversed by a low-FODMAP diet, indicating that the elevated levels of LPS contribute to IBS symptoms [94]. This finding also provides insight into the therapeutic benefits of a low-FODMAP diet for IBS patients, as it helps alleviate symptoms by addressing underlying gut imbalances [94]. Additionally, short-chain fatty acids (SCFAs)particularly butyrate, which is produced by the gut microbiota, play a crucial role in maintaining gut barrier function by promoting the expression of tight junction proteins [95]. However, the role of SCFAs in IBS pathophysiology remains unclear due to inconsistent findings regarding SCFA levels in IBS patients.

Edogawa et al. assessed the role of fecal proteases in barrier dysfunction and symptom development in IBS [96]. The results demonstrated that 40% of patients with PI-IBS exhibited elevated fecal proteolytic activity, which correlated with more severe symptoms, impaired gut barrier function, and reduced microbial diversity [96]. Elevated fecal proteolytic activity was shown to increase intestinal permeability and disrupt tight junctions, with the microbiota playing a key role in modulating these effects [96]. Beyond modulating tight junction proteins, the gut microbiota also influences the production of intestinal mucus, which forms a protective layer between the luminal contents and the epithelial cells [97]. Certain mucin-degrading bacterial species, such as *Ruminococcus gnavus*, *Ruminococcus torques*, and *Akkermansia muciniphila*, have been found to be elevated in patients with IBS [98,99,100]. The abundance of these species correlates with the severity of bowel symptoms, further implicating the microbiota in the pathogenesis of IBS.

## 8. Visceral Hypersensitivity and Gut Microbiome

As intestinal permeability increases, the resulting immune responses and mucosal inflammation contribute to the development of visceral hypersensitivity. Visceral hypersensitivity, a hallmark of IBS, refers to an exaggerated response to mechanical stimuli applied to the bowel, clinically manifesting as discomfort and pain [30]. Data indicate that the gut microbiome plays a significant role in modulating visceral hypersensitivity. Interventions such as probiotics and antibiotics have been shown to influence this condition, potentially alleviating symptoms by altering the composition and function of gut microbiota [101,102]. For instance, when germ-free rats were colonized with gut microbiota from individuals with IBS, their pain threshold to colorectal distension was significantly decreased [103]. This suggests a direct link between microbiota composition and heightened visceral sensitivity. Additionally, studies have demonstrated that certain probiotic strains, such as *Lactobacillus plantarum*, *Lactobacillus reuteri*, *Lactobacillus helveticus*, and *Bifidobacterium longum*, have beneficial effects in reducing visceral hypersensitivity [104,105,106]. Additional studies in germ-free mice showed heightened sensitivity, which was reversed by colonization with postnatal microflora [107]. In a clinical trial with 362 women with IBS, *Bifidobacterium infantis* significantly reduced pain, bloating, and improved bowel movements after four weeks, while *Lactobacillus rhamnosus* and *Lactobacillus plantarum* alleviated pain and bloating in other studies [105,107]. These probiotics are thought to modulate gut–brain communication, thereby mitigating discomfort associated with IBS.

The role of antibiotics in visceral sensitivity, particularly in the context of IBS, has garnered significant attention due to the gut microbiota’s influence on gut–brain signaling. Antibiotics, which can drastically alter the composition and diversity of gut flora, have been shown to exacerbate visceral hypersensitivity [30]. Preclinical studies demonstrate that early-life antibiotic use can induce long-lasting changes in microbial communities, leading to increased sensitivity to pain in adulthood [108]. These findings highlight the complex interplay between antibiotic treatment, microbial disruption, and the development of visceral pain, emphasizing the need to better understand how gut flora alterations impact IBS symptoms.

## 9. Gastrointestinal Motility and Gut Microbiome

Finally, the interplay between visceral hypersensitivity, immune activation, and the BGA affects gastrointestinal motility. Gastrointestinal motility is a complex process requiring the coordination of neurons, interstitial cells of Cajal, and smooth muscle cells [109,110]. Emerging evidence highlights the gut microbiome’s significant role in regulating gut motility. The enteroendocrine system regulates intestinal motility and sensory functions by releasing various neuropeptides and neurotransmitters [111]. Bacterial metabolites have the ability to promote the production of several key neuropeptides, such as neuropeptide Y, peptide YY, glucagon-like peptide-1 (GLP-1), cholecystokinin, and substance P [112,113,114]. These molecules play a pivotal role in coordinating gut motility, sensation, and overall gut–brain communication, impacting conditions like IBS. SCFAs, secondary bile acids, and indole, produced by bacteria such as *Clostridium*, *Bacteroides*, and *Ruminococcus*, are known to activate the secretion of GLP-1 from intestinal L-cells [103]. GLP-1 plays a key role in slowing down postprandial motility in the upper gastrointestinal tract, including the antrum, duodenum, and jejunum, while simultaneously promoting colonic transit [115,116]. Research by Li et al. found that patients with IBS-C exhibited reduced serum GLP-1 levels and diminished mucosal expression of its receptors [117]. The authors proposed that this deficiency of GLP-1 leads to the loss of its prokinetic effects in the colon, contributing to constipation and associated abdominal pain [117]. In animal models, administering the GLP-1 receptor agonist exendin-4 was shown to reduce stress-related defecation and sensitivity to visceral pain [118]. Furthermore, clinical trials have demonstrated that the synthetic GLP-1 analog ROSE-010 alleviated abdominal pain and enhanced colonic transit in IBS patients [117,119]. Although the precise molecular mechanisms behind these effects remain unclear, data suggest they may involve modulation of enteric neuronal activity, impacts on the integrity of tight junctions, and activation of serotonergic pathways in the colon.

The influence of the gut microbiome extends beyond neuropeptide modulation, as evidenced by studies demonstrating that fecal microbiota transplantation from healthy controls into germ-free humanized mice significantly altered gut transit and colonic contractility. This modulation was found to depend on dietary factors that influence the magnitude and direction of the effects [120]. Alterations in gut transit, such as those induced by polyethylene glycol or loperamide, also affected microbial community composition, which normalized once transit was restored [120]. Bacterial metabolites, like SCFAs and deconjugated bile salts, generate robust motor responses in both animals and humans [121]. Additionally, metabolites from aromatic amino acids, such as tryptamine, enhanced contractility by stimulating serotonin release [122]. Conversely, methane produced by bacterial fermentation slows down small intestinal transit [123], while hydrogen sulfide, produced by sulfate-reducing bacteria, inhibits muscle contractility in both the small intestine and colon through the activation of potassium channels [124]. However, the precise role of hydrogen sulfide in IBS remains unclear. Bacterial components, such as LPS from Gram-negative bacteria, can also modulate motility by promoting the survival of enteric nitrergic neurons through TLR-4 signaling [125].

## 10. Gut Microbiome-Derived Metabolic By-Products Contribution to IBS Pathology

The gut microbiota plays a crucial role in the development of IBS through its interactions with the host, leading to the production of various metabolic by-products such as bile acids, SCFAs, neurotransmitters, and other signaling molecules [126,127,128]. These metabolic products are significant in modulating gut physiology and contributing to IBS symptoms.

Specific microorganisms, including *Bacteroides*, *Clostridium*, *Lactobacillus*, *Listeria*, and *Bifidobacterium*, are critically involved in the biosynthesis of secondary bile acids in the human intestine [129]. Disruptions in bile acid concentrations can result in cytotoxic effects, such as inducing apoptosis, causing DNA damage, and contributing to the development of functional gastrointestinal disorders like IBS. In particular, patients with IBS-D show a significant imbalance in bile acid metabolism, characterized by an elevation in primary bile acids and a reduction in secondary bile acids [130]. This imbalance has been correlated with a decreased abundance of the Ruminococcaceae family in these patients, further linking bile acid dysregulation with IBS pathophysiology [130].

In addition to bile acids, SCFAs produced through the fermentation of dietary fibers by gut bacteria also play a key role. SCFAs, such as butyrate and propionic acid, help maintain gut barrier integrity by promoting the expression of tight junction proteins, and they also influence serotonin production from tryptophan [131]. Serotonin, which regulates gut motility and secretion, is synthesized in the gut either through enterochromaffin cells or direct bacterial synthesis [131,132]. Bacterial enzymes, such as tryptophan decarboxylase, convert tryptophan into tryptamine, which interacts with the 5-HT4R serotonin receptor, influencing gut motility and secretion [133]. This highlights the importance of gut microbiota in regulating serotonin metabolism, which is central to IBS pathology.

Further research has shown that gut-derived SCFAs, like propionic acid, can cross the blood–brain barrier, influencing brain function and behavior, thus connecting gut microbiome metabolic by-products with the BGA [134]. This link between the gut and brain may explain the psychological symptoms associated with IBS.

Additionally, bacterial components such as LPS and peptidoglycans, which are part of the bacterial cell wall, can activate the innate immune system through pattern recognition receptors. Mast cells, once activated, release histamine, cytokines, and other mediators that contribute to immune responses [135]. Histamine, in particular, is crucial in IBS as it increases gut permeability and enhances visceral hypersensitivity, both of which exacerbate IBS symptoms [135].

Recent studies using advanced technologies, such as proton nuclear magnetic resonance (NMR), have identified key microbial metabolites that differentiate IBS-D patients from healthy individuals [136]. These metabolites include cadaverine, putrescine, threonine, tryptophan, and phenylalanine [136]. However, analyzing microbial metabolites presents challenges due to factors such as the timing of sample collection and the degradation of these products [137].

## 11. Therapeutic Strategies

IBS patients exhibit specific microbial disturbances, including a higher Firmicutes-to-Bacteroidetes ratio, diminished levels of beneficial bacteria (e.g., *Bifidobacterium* and *Faecalibacterium*), and an overabundance of pathogenic bacteria (e.g., *Enterobacteriaceae* and *Escherichia coli*). These microbial alterations are associated with dysregulation of the immune system and increased intestinal permeability, further promoting IBS symptomatology. The interactions between gut microbiota and the immune system are pivotal. Mucosal immune activation, characterized by increased numbers of mast cells and pro-inflammatory cytokines, is related to gut microbiota imbalances. This immune response not only disrupts gut homeostasis but also perpetuates the cycle of visceral hypersensitivity and chronic inflammation seen in IBS. Moreover, advancements in microbiome research reveal the broader involvement of microbial by-products such as SCFAs, neurotransmitters, and bile acids in IBS. These metabolites influence gut motility, mucosal integrity, and immune responses, further linking microbial imbalances with the clinical symptoms of IBS. The potential therapeutic benefits of modulating the gut microbiota through probiotics, diet, and even fecal microbiota transplantation offer promising avenues for restoring gut homeostasis. Table 3 summarizes the current therapeutic approaches aimed at modulating the gut microbiome to address IBS symptoms. Each strategy targets specific microorganisms or their metabolic by-products to restore microbial balance, reduce inflammation, and improve gut barrier function, ultimately aiming to alleviate the diverse symptoms associated with IBS.

## 12. Future Research Directions and Methodological Limitations in IBS

Future research in IBS could benefit significantly from the integration of advanced omic technologies and machine learning (ML) techniques. ML, which learns patterns from data without explicit programming, is already being applied to microbiome studies, including microbiome composition analysis and identifying therapeutic targets [138]. Key ML methods, such as supervised learning, unsupervised learning, and reinforcement learning, are being explored for microbiome research, aiding in classification, discovering hidden patterns, and optimizing decision-making models [139,140]. These approaches, combined with next-generation sequencing (NGS) technologies like shotgun metagenomics, promise to offer deeper insights into the structure and function of the gut microbiome. Additionally, advanced techniques such as metatranscriptomics and metabolomics could enhance our understanding of the microbial functional pathways involved in IBS [141]. These methods provide a more detailed view of the complex interactions between the microbiome and host physiology, helping to identify new therapeutic targets essential for developing effective microbiome-directed interventions. The application of these tools could accelerate the identification of diagnostic biomarkers, improve patient risk assessment, and enhance the prediction of treatment responses in IBS.

Current IBS research faces several methodological limitations. Many studies rely on 16S rRNA gene sequencing, which only provides genus-level resolution and lacks functional insights into the microbiome. In contrast, more advanced techniques like shotgun metagenomic sequencing and RNA sequencing offer greater sensitivity, resolution, and a deeper understanding of microbial structure and function [142,143,144,145]. Additionally, most studies focus on stool samples, which may not fully represent the microbiome of other intestinal regions, such as the small intestine or mucosal layer. Research comparing fecal and mucosal microbiomes has shown only partial correlation, highlighting the need for sampling from multiple sites in the gut [146]. Moreover, while some studies assess microbiota at different time points, the majority are limited to two measurements, making it difficult to track microbiota and metabolite changes over time, especially during disease flare-ups or remission. As a result, despite the potential of the gut microbiota as a biomarker for IBS, a clear understanding of IBS-specific microbiota characteristics and reliable biomarkers remains elusive.

## 13. Conclusions

The BGA plays an essential role in how the microbiota communicates with the brain and the immune system of the host. This communication is crucial for maintaining gastrointestinal and psychological health. However, in IBS, this communication is disrupted, leading to dysbiosis, increased intestinal permeability, visceral hypersensitivity, and low-grade mucosal inflammation (Figure 2).

Despite significant progress in understanding the role of microbiome in IBS, the exact nature of microbial dysbiosis remains inconsistent across studies, suggesting that individual variability in microbiota profiles makes difficult the identification of a universal microbiome signature for IBS. Future research should unravel the complex gut–brain–immune interplay, particularly focusing on refining microbiota-based therapies. A comprehensive understanding of these interactions is critical for addressing the multifaceted contributions of the gut microbiome to IBS. In addition to bacteria, exploring the roles of non-bacterial gut populations such as fungi, archaea, and viruses in IBS pathogenesis will provide valuable insights. This broader perspective further supports the importance of personalized therapeutic strategies that target microbial composition, ultimately aiming to alleviate the wide spectrum of IBS symptoms and enhance patients’ quality of life.

## Figures and Tables

**Figure 1 ijms-25-12395-f001:**
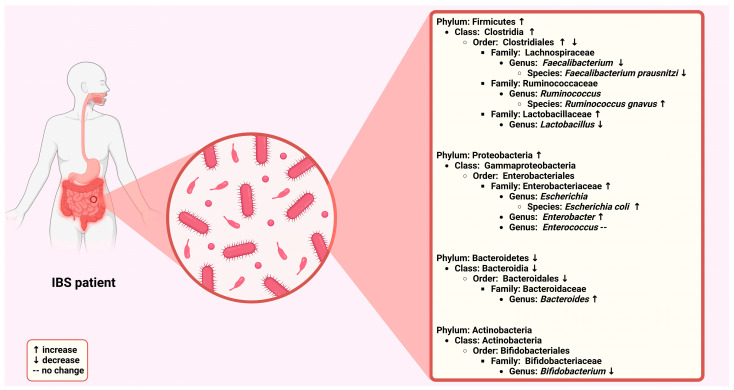
Microbiome shifts in irritable bowel syndrome. Created with BioRender.com (accessed on 14 November 2024).

**Figure 2 ijms-25-12395-f002:**
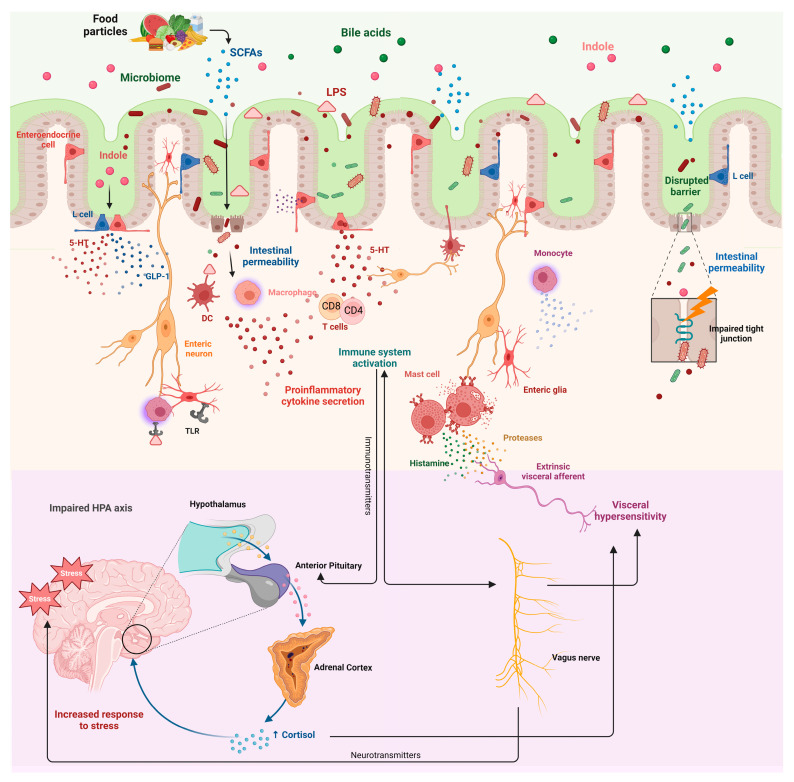
Microbiome–Gut–Brain axis interactions and their role in irritable bowel syndrome pathophysiology. Figure 2 illustrates the complex interplay between microbiome alterations, gut physiology, immune system activation, and neuroendocrine pathways in the context of irritable bowel syndrome (IBS). At the intestinal lumen, food particles are broken down, releasing various metabolites such as short-chain fatty acids (SCFAs), bile acids, and indoles. These metabolites, alongside lipopolysaccharides (LPS) from disrupted gut bacteria, modulate intestinal permeability by affecting tight junctions and enterocyte integrity. The immune system of the intestinal tissue is activated by this microbial translocation, leading to the secretion of proinflammatory cytokines by macrophages, dendritic cells (DCs), and T cells. These cytokines further perpetuate immune activation, mast cell degranulation (releasing histamine and proteases), and enteric glial cell stimulation. The activation of the immune cells and the release of neurotransmitters result in sensitization of enteric neurons, contributing to visceral hypersensitivity, a hallmark of IBS symptoms. Serotonin (5-HT), primarily released by enteroendocrine cells, alongside glucagon-like peptide-1 (GLP-1), influences gut motility and sensory signaling through enteric neurons. Impaired tight junctions allow further interactions between gut metabolites like indoles and the immune system, exacerbating gut–brain axis dysregulation. The hypothalamic–pituitary–adrenal (HPA) axis is shown to be impaired, with stress playing a critical role in increasing cortisol levels, which regulates both immune responses and the central nervous system. This feedback loop involving stress and immune mediators contributes to increased visceral hypersensitivity and IBS symptoms through vagus nerve signaling and neurotransmitter alterations. Created with BioRender.com (accessed on 1 November 2024). Abbreviations: SCFAs, short-chain fatty acids; LPS, lipopolysaccharides; 5-HT, serotonin; DC, dendritic cell; TLR, toll-like receptor; GLP-1, glucagon-like peptide-1; HPA, hypothalamic–pituitary–adrenal.

**Table 1 ijms-25-12395-t001:** Brain gut axis and gut microbiome in IBS.

Study	Microorganism	Type	Findings	Impact on BGA and IBS Pathology
[58]	*Pseudomonas aeruginosa*	Pathogenic	Growth stimulated by stress-related neurotransmitters	- Increases gut permeability - Promotes inflammation - Deteriorates visceral pain
[58]	*Campylobacter jejuni*	Pathogenic	Growth stimulated by stress-related neurotransmitters	- Increases gut permeability - Deteriorates visceral pain
[62,63]	*Bifidobacterium dentium*	Beneficial	Produces serotonin and GABA, enhances serotonin receptor expression in gut, synthesizes tyrosine	- Reduces anxiety-like behaviors - Modulates neurotransmitter levels - Supports gut–brain communication
[64]	*Bifidobacterium adolescentis*	Beneficial	Produces GABA, a neurotransmitter that influences gut–brain communication	- Modulates stress response - Supports gut homeostasis
[66]	*Bifidobacterium longum*	Beneficial	Improves depression and QoL in IBS, reduces brain response to negative stimuli	- Modulates mood and emotional response - Enhances QoL

Abbreviations: BGA, brain gut axis; IBS, irritable bowel syndrome; GABA, gamma aminobutyric acid; QoL, quality of life.

**Table 2 ijms-25-12395-t002:** Contribution of microorganisms to post-infectious IBS.

Study	Microorganism	Type	Effect on PI-IBS	Mechanisms/Findings
[71]	*Escherichia coli*	Pathogenic	Contributes to exacerbation of IBS symptoms	- Disrupts gut microbiome balance - Increases inflammation - Modifies gut motility
[71]	*Clostridium difficile*	Pathogenic	Implicated in IBS development, especially after antibiotic use	- Differentiates gut microbiota leading to dysbiosis and increasing gutpermeability
[71,72]	*Campylobacter concisus*	Pathogenic	Linked to PI-IBS	- Impairs intestinal barrier - Increases permeability - Triggers inflammatory responses
[71]	*Helicobacter pylori*	Pathogenic	Linked to PI-IBS	- Induces inflammation - Modifies gut motility
[71]	*Giardia lamblia*	Pathogenic	Commonly linked with PI-IBS, especially following gastrointestinal infection	- Disrupts gut barrier function and microbiota, triggering immune responses and inflammation
[71]	*Chlamydia trachomatis*	Pathogenic	May contribute to gut dysbiosis and PI-IBS post-infection	- Potential role in increasing gut permeability and immune system activation
[71]	*Mycobacterium avium*	Pathogenic	Possible link to IBS exacerbation post-infection	- Impairs gut immune response - Modifies microbial composition
[71]	*Salmonella* spp.	Pathogenic	Contributes to PI-IBS by causing gut permeability and inflammation	- Triggers immune response and inflammation - Alters gut motility
[71,72]	*Campylobacter jejuni*	Pathogenic	Strong association with PI-IBS	- Impairs intestinal epithelial barrier, leading to increased permeability and cellular damage
[71]	*Pseudomonas aeruginosa*	Pathogenic	May contribute to PI-IBS exacerbation	- Disrupts gut microbiota - Promotes inflammation, increasing intestinal permeability
[71]	*Shigella* spp.	Pathogenic	Can exacerbate IBS symptoms following infection	- Induces immune response - Modifies gut permeability - Disrupts motility

Abbreviations: PI-IBS, post-infection irritable bowel syndrome.

**Table 3 ijms-25-12395-t003:** Potential therapeutic approaches.

Therapeutic Approach	Mechanism of Action	Target Microorganisms	Effect on IBS Pathology
Probiotics (e.g., *Bifidobacterium longum*, *Lactobacillus plantarum*)	Modulate gut–brain axis, reduce inflammation, tighten gut barrier	Beneficial bacteria like *Bifidobacterium*, *Lactobacillus*	- Improves symptoms - Reduces visceral hypersensitivity - Enhances gut function
Prebiotics (e.g., fiber supplements)	Support development of beneficial bacteria	Increases *Bifidobacterium*, *Faecalibacterium*	- Restores microbial balance - Reduces inflammation
Antibiotics	Target pathogenic overgrowth, particularly SIBO	Pathogenic bacteria like *Escherichia coli*, *Enterobacteriaceae*	- Reduces bloating and discomfort (particularly in IBS-D)
Dietary Modifications (Low-FODMAP diet)	Reduce fermentation and inflammation	Reduces pathogenic bacteria, promoting gas production	- Decreases bloating - Improves intestinal permeability
Fecal Microbiota Transplantation	Restores gut microbial diversity	Broad spectrum (depends on donor microbiota)	- Shows promise in reducing symptoms (more research is needed)
SCFA Supplementation	Supports gut barrier function, modulates motility	SCFA-producing bacteria like *Faecalibacterium*	- Enhances gut motility and integrity - Alleviates symptoms

Abbreviations: IBS, irritable bowel syndrome; SIBO, small intestinal bacterial overgrowth; SCFA, short-chain fatty acids.

## Data Availability

No new data were created or analyzed in this study. Data sharing is not applicable to this article.

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
