# Peer review of "Microbiome Shifts and Their Impact on Gut Physiology in Irritable Bowel Syndrome"

_ijms, 2024, doi:10.3390/ijms252212395_

Round 1

Reviewer 1 Report

Comments and Suggestions for Authors

The manuscript entitled Microbiome Shifts and Their Impact on Gut Physiology in Irritable Bowel Syndrome is a high-quality review paper that covers in detail all aspects of the relationship between the microbiome and irritable bowel syndrome, and, as such represents a significant contribution to this scientific field.

In this sense, I have no objections and propose publication without further changes.

Author Response

Reviewer 1

The manuscript entitled Microbiome Shifts and Their Impact on Gut Physiology in Irritable Bowel Syndrome is a high-quality review paper that covers in detail all aspects of the relationship between the microbiome and irritable bowel syndrome, and, as such represents a significant contribution to this scientific field.

In this sense, I have no objections and propose publication without further changes.

Response to Reviewer 1:   Thank you very much for your positive feedback on our manuscript. We greatly appreciate your kind words and your support for the publication of our work without further changes.

Reviewer 2 Report

Comments and Suggestions for Authors

In this review manuscript, the authors systematically summarize the variations in the microbiome and their influence on the gut and the microbiome-gut-brain axis in irritable bowel syndrome (IBS). They specifically focus on the impact of various bacteria, including the increase in beneficial species (e.g., Lactobacillus and Bifidobacterium) and the decrease in potentially harmful species (e.g., Firmicutes and Proteobacteria). Additionally, they provide detailed information on microbiome-derived influences, covering areas such as IBS classification, current IBS status, potential mechanisms of IBS, the microbiome-gut-brain axis, post-infectious IBS, immune response, intestinal permeability, visceral hypersensitivity, gastrointestinal motility, and metabolic by-products.

In summary, this review presents a comprehensive and novel perspective on the interactions between the microbiome and IBS, providing a valuable and accessible introduction for readers interested in IBS-related research. However, a few points in the manuscript required attention and revision. I suggest the paper be considered for publication in the International Journal of Molecular Sciences after addressing the following concerns and suggestions.

1. Main text readability and structure: The main text introducing IBS and related information is comprehensive and valuable. However, it is difficult for me (and possibly other readers) to quickly and easily identify the key points and major conclusions of each subpart. This suggests that the text description alone may lack readability. I recommend that the authors incorporate additional presentation ways, such as Tables and Figures, to summarize the essential points of each subpart (at the authors’ discretion). The following papers (doi) may provide useful references for this suggestion:

10.3390/microorganisms11102369

10.3389/fcimb.2020.00468

10.3389/fcimb.2021.729346

10.3390/jcm12072558

10.3390/microorganisms11082089

2. Highlighting the major focus of the review: As noted above, many review articles on IBS and the microbiome have been published in recent years. I suggest that the authors clearly highlight the unique focus of their review compared to similar papers. Based on the manuscript’s title, it seems that this unique focus may be on “microbiome shifts”. However, this point is not emphasized clearly, possibly due to a lack of straightforward Tables or Figures. I recommend the authors to revise for highlighting this focus.

3. Figure clarity and readability: The Figure in this manuscript contains substantial information, making it comprehensive. However, the schematic diagram is somewhat unclear (low resolution) and the font size is small. I suggest the authors to enlarge the diagram and text in the revised manuscript. Additionally, if the Figure was created using commercial software or an online tool, this should be noted and acknowledged in the manuscript.

Author Response

Reviewer 2

In this review manuscript, the authors systematically summarize the variations in the microbiome and their influence on the gut and the microbiome-gut-brain axis in irritable bowel syndrome (IBS). They specifically focus on the impact of various bacteria, including the increase in beneficial species (e.g., Lactobacillus and Bifidobacterium) and the decrease in potentially harmful species (e.g., Firmicutes and Proteobacteria). Additionally, they provide detailed information on microbiome-derived influences, covering areas such as IBS classification, current IBS status, potential mechanisms of IBS, the microbiome-gut-brain axis, post-infectious IBS, immune response, intestinal permeability, visceral hypersensitivity, gastrointestinal motility, and metabolic by-products.

In summary, this review presents a comprehensive and novel perspective on the interactions between the microbiome and IBS, providing a valuable and accessible introduction for readers interested in IBS-related research. However, a few points in the manuscript required attention and revision. I suggest the paper be considered for publication in the International Journal of Molecular Sciences after addressing the following concerns and suggestions.

Comment 1: Main text readability and structure: The main text introducing IBS and related information is comprehensive and valuable. However, it is difficult for me (and possibly other readers) to quickly and easily identify the key points and major conclusions of each subpart. This suggests that the text description alone may lack readability. I recommend that the authors incorporate additional presentation ways, such as Tables and Figures, to summarize the essential points of each subpart (at the authors’ discretion). The following papers (doi) may provide useful references for this suggestion:

10.3390/microorganisms11102369

10.3389/fcimb.2020.00468

10.3389/fcimb.2021.729346

10.3390/jcm12072558

10.3390/microorganisms11082089

Response to comment 1:  To address this issue, we have incorporated additional presentation ways in the revised version. Specifically, we have added a new figure illustrating the microbial shifts observed in IBS (figure 1), as well as two tables: one summarizing the role of the gut microbiome in the brain-gut axis (BGA) (table 1) and another describing the influence of specific microorganisms on IBS development, particularly in the context of post-infectious IBS (PI-IBS) (table 2). These additions aim to provide a clear and concise overview of the key points and major conclusions in each subpart, improving the accessibility of the text for readers.

Comment 2: Highlighting the major focus of the review: As noted above, many review articles on IBS and the microbiome have been published in recent years. I suggest that the authors clearly highlight the unique focus of their review compared to similar papers. Based on the manuscript’s title, it seems that this unique focus may be on “microbiome shifts”. However, this point is not emphasized clearly, possibly due to a lack of straightforward Tables or Figures. I recommend the authors to revise for highlighting this focus.

Response to comment 2: We agree that many reviews on IBS and the microbiome have been published recently, and we appreciate the opportunity to clarify the unique focus of our work. In the revised manuscript we have emphasized on the role of our review which was the concept of “microbiome shifts in IBS, focusing on dynamic alterations in the gut microbial composition, including specific taxa that are either enriched or depleted, as well as the resulting physiological impacts on IBS pathology (p.1, lines 17-19 and p. 2-3, lines 77-85). In addition, to better distinguish our review, we have incorporated additional two tables and 1 figure, as mentioned in the previous comment, that summarize findings related to these microbiome shifts and their connections to key IBS-related processes (e.g., intestinal permeability, visceral hypersensitivity, and immune activation).

Comment 3: Figure clarity and readability: The Figure in this manuscript contains substantial information, making it comprehensive. However, the schematic diagram is somewhat unclear (low resolution) and the font size is small. I suggest the authors enlarge the diagram and text in the revised manuscript. Additionally, if the Figure was created using commercial software or an online tool, this should be noted and acknowledged in the manuscript.

Response to comment 3:   Thank you for your valuable feedback. In response, we have enlarged the font in the schematic diagram to ensure it is more clearly visible. Regarding the resolution, we have submitted a high-resolution version of the figure to the journal. Additionally, as per your suggestion, we have now included information about the software platform used to create the figure in the revised manuscript.  

Reviewer 3 Report

Comments and Suggestions for Authors

This article comprehensively reviews the literature on gut microbiota alterations that impact IBS, focusing on crucial aspects such as the brain-gut axis, immune activation, gut permeability, and motility. The paper accurately represents the microbiome's influence on IBS symptomatology and discusses various therapeutic approaches. Additionally, it highlights the roles of cytokines and the gut-brain axis.

To enhance clarity and readability, it is recommended to add 2-3 tables that summarize:
1. The findings of each study related to specific microorganisms, both pathogenic and beneficial.
2. A consolidated table of potential therapeutic approaches.

Lastly, a brief section on future research directions in IBS, such as omic technologies and next-generation sequencing approaches, along with the methodological limitations of the current research, should be added before the Conclusion.

Author Response

Reviewer 3

This article comprehensively reviews the literature on gut microbiota alterations that impact IBS, focusing on crucial aspects such as the brain-gut axis, immune activation, gut permeability, and motility. The paper accurately represents the microbiome's influence on IBS symptomatology and discusses various therapeutic approaches. Additionally, it highlights the roles of cytokines and the gut-brain axis.

Comment 1: To enhance clarity and readability, it is recommended to add 2-3 tables that summarize the findings of each study related to specific microorganisms, both pathogenic and beneficial and a consolidated table of potential therapeutic approaches.

Response to comment 1:  To address this issue, we have added a new figure illustrating the microbial shifts observed in IBS (figure 1), as well as two tables: one summarizing the role of the gut microbiome in the brain-gut axis (BGA) (table 1) and another describing the influence of specific microorganisms on IBS development, particularly in the context of post-infectious IBS (PI-IBS) (table 2). In parallel, we have added a section titled “Therapeutic Strategies” accompanied by a table (Table 3) which presents the current therapeutic approaches aimed at modulating the gut microbiome to address IBS symptoms These additions aim to provide a clear and concise overview of the key points and major conclusions in each subpart, improving the accessibility of the text for readers.

Comment 2: Lastly, a brief section on future research directions in IBS, such as omic technologies and next-generation sequencing approaches, along with the methodological limitations of the current research, should be added before the Conclusion.

Response to comment 2:  In the revised manuscript we have added a section titled “Future Research Directions and Methodological Limitations in IBS”, as suggested by the reviewer (p. 15-16, lines 487-517).